# LDStega: Practical and Robust Generative Image Steganography based on Latent Diffusion Models

## ABSTRACT

Generative image steganography has gained significant attention due to its ability to hide secret data during image generation. However, existing generative image steganography methods still face challenges in terms of controllability, usability, and robustness, making it difficult to apply real-world scenarios. To ensure secure and reliable communication, we propose a practical and robust generative image steganography based on Latent Diffusion Models, called LDStega. LDStega takes controllable condition text as input and designs an encoding strategy in the reverse process of the Latent Diffusion Models to couple latent space generation with data hiding. The encoding strategy selects a sampling interval from a candidate pool of truncated Gaussian distributions guided by secret data to generate the stego latent space. Subsequently, the stego latent space is fed into the Decoder to generate the stego image. The receiver extracts the secret data from the globally Gaussian distribution of the lossy-reconstructed latent space in the reverse process. Experimental results demonstrate that LDStega achieves high extraction accuracy while controllably generating image content and saving the stego image in the widely used PNG and JPEG formats. Additionally, LDStega outperforms state-of-the-art techniques in resisting common image attacks.

## CCS CONCEPTS

• **Information systems** → *Multimedia information systems*; • **Security and privacy** → *Security services*.

## KEYWORDS

Image steganography, Latent diffusion model, Date hiding

## 1 INTRODUCTION

The rapid advancements in AI-generated content (AIGC) have led to significant concerns regarding data privacy, security, and protection. Image steganography is a technique used for covert communication, where secret data are concealed within images to prevent unauthorized access or detection. Traditional image steganography involves modifying the cover image to embed secret data, including both hand-crafted and deep learning-based methods. However, these methods often leave explicit traces of the secret data as artifacts or local details in the stego image, making them susceptible to

*ACM MM, 2024, Melbourne, Australia*

© 2024 Copyright held by the owner/author(s). Publication rights licensed to ACM.
ACM ISBN 978-x-xxxx-xxxx-x/YY/MM
https://doi.org/10.1145/nnnnnnn.nnnnnnn

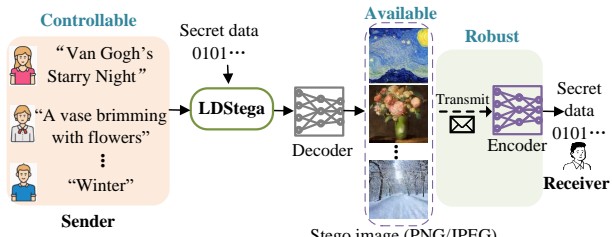

**Figure 1: LDStega excels at tailoring stego images to align with the sender's characteristics and secret data. Stego images are saved in popular PNG or JPEG formats and transmitted through lossy channels. The secret data can be recovered from received stego images by an Encoder of the LDM.**

detection by well-designed steganalysis techniques, reducing the security of steganography.

Recently, significant progress has been made in image generation using generative models, producing the proposal of generative image steganography (GIS) [19, 23, 32, 33, 41]. While GIS shows promising performance in resisting typical steganalysis attacks, there are still certain drawbacks that need to be addressed. GAN-based DCGAN-Steg [13] is limited by the choice of model, resulting in generated images with insufficient fidelity. Although S2IRT [41] based on the Glow model and IDEAS [19] based on image disentanglement autoencoder improve the quality of generated images, they are trained in a noiseless simulation environment, making them vulnerable to real-world attacks. Additionally, these methods save stego images as floating-point types instead of integer types, which severely impacts their robustness and usability. RoSteALS [4] and CtrGAN [40] consider robustness but overlook the problem of quantization rounding of stego images. Furthermore, the generating content of the stego images in these methods lacks controllability, as generated images are only randomly sampled from the generation model. For a steganographer, it's essential to take into account both the resistance to steganalysis of the steganographic image and its covert behavior, that is, the stego image needs to be tailored according to the sender's characteristics, including his/her occupation and interests, thereby preventing any suspicion from arising due to unusual behavior. Overall, the current methods lack a comprehensive solution that encompasses controllability, availability, and robustness.

Recently, diffusion models, especially Latent Diffusion Models (LDM), have developed a lot, facilitating text-based conditional image generation, which aligns well with our steganography task's need for controllability. Meanwhile, large-scale LDM communities have contributed an extensive collection of freely available open-source tools, which provides a good camouflage environment for steganography. However, how to couple message hiding and image generation in LDM, while the process is also robust to lossy data

storage and lossy channel transmission operations, remains a problem. To address this, we propose LDStega, a practical and robust generative image steganography based on LDM, as illustrated in Fig. 1. Specifically, LDStega conceals secret data within the noise and utilizes a deterministic sampler to generate the latent space of the stego image. Subsequently, a Decoder is then used to generate the stego image from this latent space. However, the discretization, noise addition, and encoding processes introduce information loss in the latent space, leading to diminished decoding accuracy of the secret data. To enhance the precision of secret data recovery, a mapping function based on a truncated Gaussian distribution is designed. For the receiver, the shared random seed and condition can be utilized to replicate the generation process of the latent space, thereby obtaining the probability distribution of the latent space and facilitating secret data extraction. The main contributions of LDStega can be summarized as follows:

- **Latent steganography:** By experimentally exploring three key properties of LDM, we design practical and robust generative image steganography based on LDM. LDStega conceals secret data within the noise and utilizes a deterministic sampler to generate the stego latent space. Notably, LDStega does not require fine-tuning pre-trained models or training additional models.

- **Practicality:** LDStega utilizes the controlled condition text as input, enabling the sender to generate personalized stego images for various scenarios with the sender's characteristics, thereby avoiding suspicion arising from unusual behavior. Furthermore, LDStega saves stego images in the commonly used PNG and JPEG formats, effectively addressing the issue of reduced extraction accuracy resulting from quantization errors.

- **Robustness:** We design a coding strategy that selects a sampling interval from a candidate pool of truncated Gaussian distributions guided by secret data to generate the stego latent space, which ensures high extraction accuracy in resisting common image attacks.

## 2 PRELIMINARIES AND RELATED WORK

### 2.1 Steganography based on embedding

For steganography with embedding, the earliest traditional methods embed secret data based on least significant bits (LSBs) replacement [30]. After that, adaptive steganography [8, 11, 12, 17, 18, 31, 39] is proposed to find suitable regions to modify during the embedding process. In recent years, deep learning-based techniques have been used for steganographic tasks [2, 9, 16, 20] because of their powerful learning capabilities. Baluja et al. [2] proposed an encoder and a decoder for concealing and extracting secret color images, respectively. FNNS [16] exploits neural networks' sensitivity to tiny perturbations, achieving a reliable 0% error rate when concealing up to 3 bits per pixel (bpp) of secret data in images. To improve security, AdaSteg [22] utilized deep reinforcement learning and encrypted noises to implement adaptive local image steganography. Following, invertible neural networks (INNs) [9, 14, 20, 34] are used to implement large-capacity steganography. Nevertheless, steganography with embedding has an inherent risk that modification traces of the cover image are inevitably left. This, in turn, may result in easy detection of the stego image by well-designed steganalysis techniques [3, 35, 36].

## 2.2 Generative image steganography

Generative image steganography utilizes generative models to hide secret data during image generation. In 2022, Liu et al. [19] proposed an image disentanglement autoencoder for steganography (IDEAS), which exploits the structure representation's stability to improve the secret data's decoding rate. However, it suffers from the inefficiency and irreversibility of the secret-to-image transformation. To address this issue, Zhou et al. [41] proposed an S2IRT scheme that utilizes the Glow model to establish a bijective mapping between the latent space with a multivariate Gaussian distribution and the image space with a complex distribution. However, S2IRT has limitations regarding the visual quality and diversity of stego images. CtrGAN [40] introduces a generative steganographic framework that auto-generates semantic object contours. It encodes the given secret data as these object contours, preserving their distribution for stego image generation. RoSteALS [4] proposes a practical steganography technique leveraging frozen pretrained autoencoders to free the payload embedding from learning the distribution of cover images. Additionally, the stego images of these methods are saved as floating-point types rather than integer types, which severely impacts their availability in the real world. GSN [32] integrates a mutual information mechanism for synthesizing stego images. It consists of four sub-networks: a discriminator, a steganalyzer, an extractor, and a generator. While GSN [32] addresses the issue of quantization error, it is still trained in a noiseless simulation environment, failing to improve the robustness of steganography.

### 2.3 Diffusion models

Due to its powerful network representation, diffusion models, which are trained to model the target image distribution starting from a noise distribution, have been applied to a wide variety of generative modeling tasks, such as image generation [27, 29], image inpainting [6, 7, 21], image editing [1, 5], among others. Diffusion models involve a forward diffusion process where noise is progressively added to the image to create a noisy image conforming to a Gaussian distribution, and a backward image generation process where the noisy image is gradually denoised to yield a natural image. The denoising diffusion probabilistic model (DDPM) [5, 10] has multiple image generation steps and operates directly in the pixel space, resulting in long processing times and high inference costs. Fortunately, a recent advancement called Latent Diffusion Models [24] has applied the diffusion process and its inverse to the latent space of images. This improvement significantly reduces training costs and greatly enhances image fidelity compared to previous diffusion models. Recently, Yu et al. [38] utilize the image transformation capability of LDM to realize the conversion of secret image to container image. Unlike their approach, which hides the secret image into the container image, LDStega designs an encoding strategy in the reverse process of the LDM to hide binary bit stream to improve extraction accuracy of secret data and enhance universality.

## 3 PROPOSED APPROACH

In this paper, we propose a practical and robust generative image steganography based on LDM to achieve controllability, availability, and robustness in secret data hiding. In the LDStega framework, illustrated in Fig. 2, the sender can personalize the stego image by

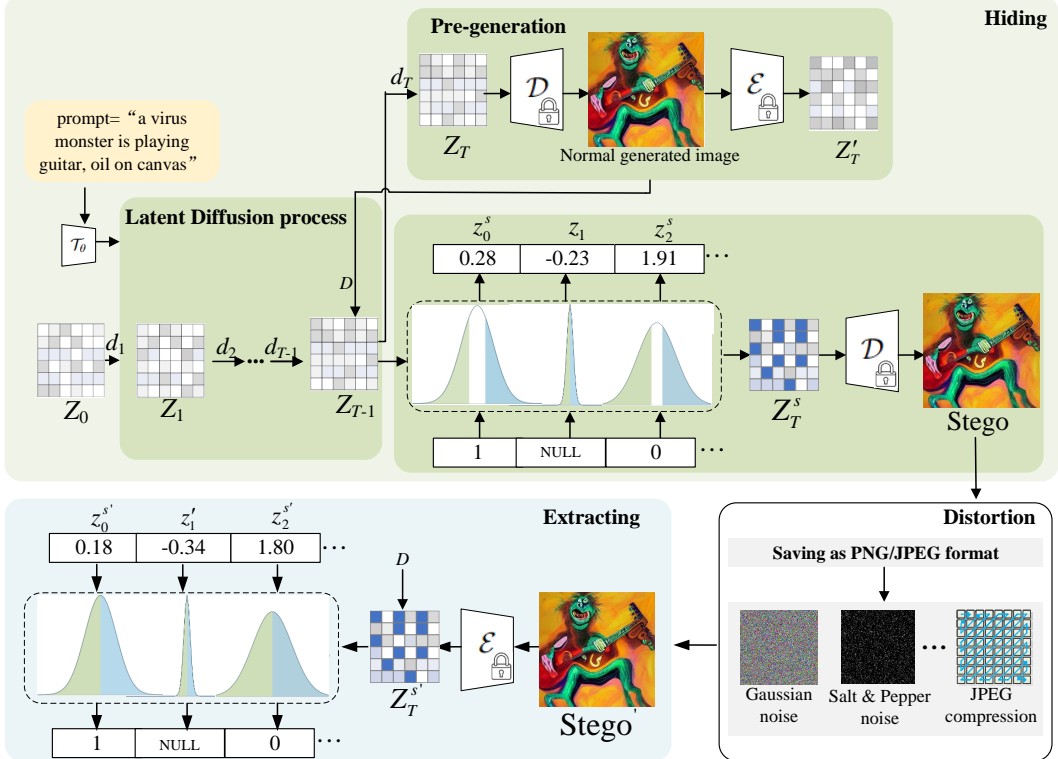

**Figure 2: The framework of LDStega. Both parties (Sender and Receiver) need to share the same settings: the random seed and the pretrained LDM. In the hiding process, the sender takes controllable condition text as input and maps the secret data in the latent space $Z_T^s$. Then, LDStega generates the stego image $X^s = \mathcal{D}\left(Z_T^s\right)$ and saves $X^s$ as PNG or JPEG formats. At the receiver's end, the stego image $X^r = Robust\left(X^q\right)$ is received over the lossy channel. In the extraction process, the receiver can synchronize all states with the sender and extract the message from the stego image $X^r$.**

transforming input text into text embedding through a Condition Encoder $\mathcal{T}_\theta$. By leveraging the pretrained LDM's characteristics, which exhibit the loss of reconstructing the latent space in the diffusion process and robustness of image reconstruction in the reverse process, LDStega designs an encoding strategy utilizing the probability distribution of the latent space to hide secret data. The sender uses this encoding strategy of the truncated Gaussian distribution to encode the secret data into the latent space $Z_T^s$. Following this, the stego latent space is fed into the Decoder to generate the stego image. Stego images are stored in widely used PNG or JPEG formats and transmitted through lossy channels. In the extracting process, the receiver extracts the secret data from the globally Gaussian distribution of the lossy-reconstructed latent space by executing the inverse procedure of message hiding.

## 3.1 Leveraging pretrained Latent Diffusion Models

To explore whether a pretrained LDM [25] has steganographic capabilities, LDStega conducts experiments using an LDM that accepts condition text input to control the content of the generated image. Taking Fig. 2 as an illustration, the condition text (prompt="a virus monster is playing guitar, oil on canvas") is initially inputted into

the Condition Encoder $\mathcal{T}_\theta$ to derive the corresponding condition embedding $cd$. Simultaneously, a latent space $Z_0$ of size $H^{'} \times W^{'} \times C^{'}$ is sampled from a Gaussian distribution. Following this, both $Z_0$ and the condition embedding $cd$ are fed into the conditional inverse process of the denoising diffusion implicit model (DDIM). The process can be mathematically represented as follows:

$$Z_T = ODESolve(Z_0; \epsilon_\theta, cd, 0, T), \qquad (1)$$

where $ODESolve$ is an Ordinary Differential Equation (ODE) solver [26], and $\epsilon_\theta$ represents a pretrained noise estimator. After $T$ steps of diffusion, the obtaining $Z_T$ is inputted a Decoder $\mathcal{D}$ to produce an image $X$ of size $H \times W \times C$, where $H^{'} \leq H$ and $W^{'} \leq W$. In the process of reconstructing the latent space, the pretrained Encoder $\mathcal{E}$ is employed to reconstruct the latent space $Z_T^{'}$ from the generated image $X$.

$$Z_T^{'} = \mathcal{E}(X). \qquad (2)$$

By reconstructing $Z_T^{'}$ of the generated image $X$ and applying quantization and noise to $X$, we draw three conclusions:

**(i) The reconstructed $Z_T^{'}$ experiences a loss in comparison to $Z_T$, and similarly, $X^{'} = \mathcal{D}\left(\mathcal{E}(X)\right)$ incurs a loss compared to $X$.** As shown in Fig. 3, the leftmost column displays images

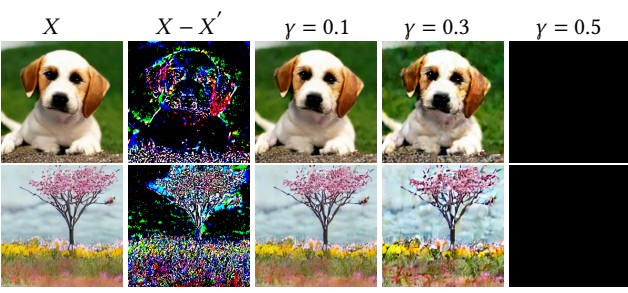

**Figure 3: Correlation between sampling intervals in the latent space and the generated image, where the size of the truncated intervals $\gamma$ is set to 0.1, 0.3, and 0.5, respectively.**

$X$ generated from the condition text "a dog". The second column illustrates the residual between the image $X$ and $X'$ generated using the reconstructed $Z_T'$.

**(ii) After applying quantization and noise layer to the generated image $X$, the discrepancy between each element of the reconstructed $Z_T'$ and $Z_T$ remains mostly within 0.3.** As shown in Fig. 4, where the latent space of size 32×32×4, we count the number of elements in discrepancy $D = Z_T' - Z_T$ that fall in each of these six intervals $MS_0 = [0, 0.05]$, $MS_1 = (0.05, 0.1]$, $MS_2 = (0.1, 0.15]$, $MS_3 = (0.15, 0.2]$, $MS_4 = (0.2, 0.25]$, $MS_5 = (0.25, 0.3]$. There are 1,636 elements in $D$ within the range of $MS_0$, followed by 1,169 elements, and so on, down to 71 elements within the $MS_5$ category. Additionally, by generating $Z_T''$ with a truncated Gaussian distribution for elements at targeting positions in $Z_T$ where $D$ exhibits relatively low values, we find that $D'' = Z_T'' - \mathcal{E}\left(\mathcal{D}(Z_T'')\right)$ also are comparatively low for elements at its corresponding positions.

**(iii) The sampling process of $Z_T$ obeys the Gaussian distribution, and within a range of truncation intervals, the difference between images generated by global Gaussian sampling and truncated Gaussian sampling is minimal.** As illustrated in the rightmost three columns of Fig. 3, the images are generated with truncated intervals $\gamma$ on both the left and right sides of

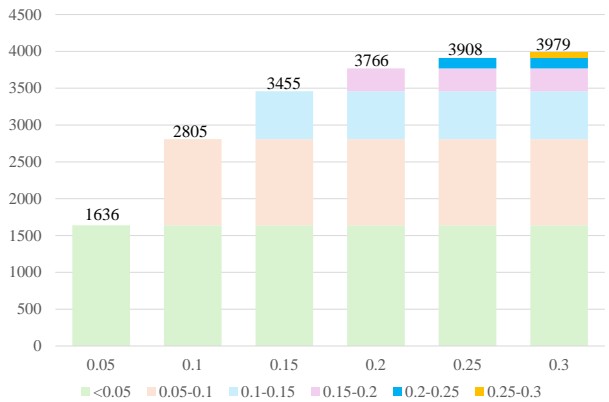

**Figure 4: With a latent space size of $32 \times 32 \times 4$, we count the number of elements in $D$ that fall in each of these six intervals.**

the symmetry axis of the Gaussian distribution probability density function set to 0.1, 0.3, and 0.5, respectively. It is noticeable that setting $\gamma$ to 0.1 and 0.3 has a minor impact on the generated image, whereas when $\gamma = 0.5$, the impact becomes substantial due to the significant deviation of the sampled value from the mean of the Gaussian distribution $\mu_{T-1}$. Based on both (i) and (ii), it is apparent that the reconstruction of the latent space yields a loss in all three cases when the generated image remains unprocessed, quantized, and noisy. Regarding (iii), when $\gamma \leq 0.3$, the generated image is insensitive to this variation.

$$Z_T'' = Samping\left((-\infty, \mu_{T-1} - \gamma), (\mu_{T-1} + \gamma, +\infty)\right), \quad (3)$$

$$\mathcal{D}\left(Z_T''\right) \approx X, \quad (4)$$

where $Samping(\cdot)$ denotes the sampling function associated with the Gaussian distribution. Therefore, the mapping function designed by LDStega should be able to robustly hide the secret data in the generation process from $Z_{T-1}$ to $Z_T$ in the LDM.

## 3.2 Steganography network based on Latent Diffusion Models

LDStega requires three pretrained submodels of the LDM: a Conditional Encoder $\mathcal{T}_\theta$, a Decoder $\mathcal{D}$, and an Encoder $\mathcal{E}$. The communication process involves two participants: the sender and receiver, with the stego image transmitted through a lossy channel. For generating stego images, LDStega initiates by initializing the latent space $Z_0$ using a random seed, ensuring $Z_0 \sim \mathcal{N}(0, \mathbf{I})$, where $\mathbf{I}$ denotes the identity matrix. The condition text is encoded into $cd$ using the conditional encoder $\mathcal{T}_\theta$.

$$cd = \mathcal{T}_\theta(Text). \quad (5)$$

Subsequently, $Z_0$ and $cd$ are utilized as inputs for the first step of the DDIM inverse process, resulting in the derivation of the mean $\mu_0$ and the variance $\sigma_0$:

$$(\mu_0, \sigma_0) = f(Z_0, cd), \quad (6)$$

where $f(\cdot)$ represents the DDIM network. Following Eq. (7), the latent space $Z_1$ is obtained, and then $Z_1$ continues to be fed into the network to get $\mu_1$ and $\sigma_1$. This process is repeated until $Z_{T-1}$ is generated, where $N_t \sim \mathcal{N}(0, \mathbf{I})$.

$$Z_{t+1} = \mu_t + \sigma_t \cdot N_t. \quad (7)$$

To minimize secret data loss caused by the reconstruction of the latent space and the lossy transmission of stego image, LDStega designs a robust mapping function $H(\cdot)$ to hide the secret data into the latent space $Z_T^s$ during the diffusion process from $Z_{T-1}$ to $Z_T$. Given that both the sender and receiver share the same random seeds, allowing the model to reverse the LDM process and reproduce $X_T$, the sender can perform pre-generation $X = \mathcal{D}(Z_T)$. Subsequently, the sender utilizes the Encoder to reconstruct an approximation of the latent space, denoted as $Z_T' = \mathcal{E}(X)$. Finally, the sender calculates the discrepancy $D = Z_T - Z_T'$. According to Section 3.1 (ii) property, LDStega divides the range of values of $D$ into seven intervals $MS = \{MS_0, MS_1, \ldots, MS_6\}$, where $MS_6 = (0.3, +\infty)$. When the steganographic capacity is less than $H' \times W' \times$

$C'$, we employ Eq. (8) to hide secret data in targeting positions where $D$ exhibits relatively low values during the diffusion process from $Z_{T-1}$ to $Z_T$, until the entire secret data is effectively concealed.

$$\begin{cases} Z_T^s[i] = H\left(m_k, \mathcal{N}\left(\mu_{T-1}[i], \sigma_{T-1}^2[i]\right), \gamma\right), & if D[i] \in MS_u \\ Z_T^s[i] = \mu_{T-1}[i] + \sigma_{T-1}[i] \cdot N_{T-1}[i], & if D[i] \notin MS_u \end{cases} \quad (8)$$

where $u \in \{0, 1, \cdots, 6\}$, $i \in \left\{1, 2, \cdots, H' \times W' \times C'\right\}$. Following the completion of diffusion, the Decoder $\mathcal{D}$ is utilized to generate the stego image $X^s$.

$$X^s = \mathcal{D}\left(Z_T^s\right). \quad (9)$$

While Wei et al. [33] showed that saving stego images in floating-point TIFF format in superior extraction accuracy compared to integer-based PNG or JPEG formats, it's noteworthy that the TIFF format occupies more storage space than the PNG and JPEG formats. Additionally, PNG and JPEG formats are more prevalent on platforms such as social media. Consequently, LDStega stores $X^s$ as an integer type PNG or JPEG formatted image $X^q$.

$$X^q = Quant\left(X^s\right), \quad (10)$$

where $Quant(\cdot)$ denotes the quantitative function applied to store the stego image in PNG or JPEG format. At the receiver's end, the stego image $X^r$ is received over the lossy channel.

$$X^r = Robust\left(X^q\right), \quad (11)$$

where $Robust(\cdot)$ represents the noise layer applied to the stego image in PNG or JPEG format. It is worth noting that both the sender and receiver must share the pretrained LDM and mapping function. The detailed procedure for generating the stego image $X^r$ using LDStega is outlined in Algorithm 1.

## 3.3 Message hiding and extraction

3.3.1 *Message hiding.* LDStega conceals the secret data in the generated image through four primary steps: preprocessing secret data, designing the mapping function $H(\cdot)$, constructing of stego latent space $Z_T^s$, and generating stego image $X^s$. In the preprocessing stage, the sender encrypts the secret data $m$ of length $l$ using the key $k_1$ to enhance security, resulting in uniformly distributed encrypted secret message $m^e$. Then, to ensure robust extraction of secret data while maintaining the quality of generated images, LDStega leverages the Gaussian distribution property of $Z_T$ to hide the encrypted data $m^e$. It designs candidate pools $pool_k = \left\{pool_{k,0}, pool_{k,1}\right\}$, symmetrically positioned about the mean $\mu_{T-1}$ of the Gaussian distribution's probability density function, for $k$ in the set $k \in \{1, 2, \cdots, l\}$. Then, driven by the secret data, one candidate pool is selected as the sampling interval. To enhance the robustness of LDStega, a parameter with a truncated interval $\gamma$ is introduced. This modification confines the two candidate pool intervals to the ranges $(-\infty, \mu_{T-1} - \gamma)$ and $(\mu_{T-1} + \gamma, +\infty)$, respectively. To further enhance security, LDStega encrypts $pool_k$ using the key $k_2$, yielding encrypted candidate pools labeled $Ind[0] = (c[0][0], c[0][1])$ and $Ind[1] = (c[1][0], c[1][1])$ from left to right. When the secret data are 0, the candidate pool $Ind[0]$ is sampled using the truncated Gaussian distribution. Conversely, when the secret data

---

**Algorithm 1** Generate stego image $X^r$

**Require:** $f$, $\mathcal{D}$ and $\mathcal{E}$ are the diffusion process, Decoder and Encoder of the pretrained LDM, respectively. The diffusion steps $T$, seed $Seed = \{d_0, d_1, \cdots, d_T\}$, secret data $m$. $H(\cdot)$, $Sampling(\cdot)$, $Quant(\cdot)$, $Robust(\cdot)$, $MS = \{MS_0, MS_1, \cdots, MS_6\}$, $\gamma$, and calculating length function $Len(\cdot)$.

**Eusure:** $X^r$
1: $l = Len(m)$, $k = 0$
2: $cd = \mathcal{T}_\theta(Text)$
3: **for** $t \in \{0, 1, \cdots, T\}$ **do**
4:     **if** $t = 0$ **then**
5:         $Z_0 = Sampling(d_0, \mathcal{N}(0, \mathbf{I}))$
6:     **else**
7:         $(\mu_t, \sigma_t) = f(Z_t, cd)$
8:         $N_t = Sampling(d_t, \mathcal{N}(0, \mathbf{I}))$
9:         $Z_{t+1} = \mu_t + \sigma_t \cdot N_t$
10:     **end if**
11:     $X = \mathcal{D}(Z_T)$, $Z_T' = \mathcal{E}(X)$, $D = Z_T - Z_T'$
12:     **if** $k < l$ **then**
13:         **for** $u \in \{0, 1, \cdots, 6\}$ **do**
14:             **for** $i \in \left\{1, 2, \cdots, H' \times W' \times C'\right\}$ **do**
15:                 **if** $D[i] \in MS_u$ **then**
16:                     $Z_T^s[i] = H\left(m_k, \mathcal{N}\left(\mu_{T-1}[i], \sigma_{T-1}^2[i]\right), \gamma_u\right)$
17:                     $k = k + 1$
18:                 **else**
19:                     $Z_T^s[i] = \mu_{T-1}[i] + \sigma_{T-1}[i] \cdot N_{T-1}[i]$
20:                 **end if**
21:             **end for**
22:         **end for**
23:     **end if**
24:     $X^s = \mathcal{D}\left(Z_T^s\right)$, $X^q = Quant(X^s)$, $X^r = Robust(X^q)$
25: **end for**

---

are 1, the candidate pool of $Ind[1]$ is sampled using the truncated Gaussian distribution.

$$z_i^s = trunc\left(\mu_{T-1}^i, \sigma_{T-1}^i, \left(c\left[m_k^e\right][0], c\left[m_k^e\right][1]\right)\right). \quad (12)$$

This process is repeated until all the secret data are concealed, and generating the stego latent space $Z_T^s$. Subsequently, Eq. (9) is employed to produce the stego image $X^s$. The detailed procedure of the mapping function $H(\cdot)$ is illustrated in Algorithm 2.

3.3.2 *Message extraction.* In the secret data extraction phase, the receiver initially obtains $Z_T^{s'}$ from the received $X^r$ using the Encoder $\mathcal{E}$. The generation process of $Z_{T-1}$ is then reproduced utilizing the shared random seed $Seed$ and condition text $cd$. The mean $\mu_{T-1}$ and standard deviation $\delta_{T-1}$ is computed via Eq. (6). The receiver sequentially extracts the secret data from $Z_T^{s'}$ based on $D$ and the length $l$ of the shared secret message. This extraction sequence is performed according to seven interval in $MS = \{MS_0, MS_1, \ldots, MS_6\}$. The receiver reconstructs the candidate pools $pool_{k,0}$ and $pool_{k,1}$ with the mean $\mu_{T-1}$ of the probability density function of Gaussian distribution as the axis of symmetry. Their intervals are $(-\infty, \mu_{T-1}]$ and $(\mu_{T-1}, +\infty)$. To maintain the same labeling order as the sender, LDStega encrypts the candidate pools

**Algorithm 2** Mapping function $H(\cdot)$

---

**Require:** $\mu_{T-1}, \sigma_{T-1}$, the secret message $m$, truncation factor $\gamma$, keys $k_1$ and $k_2$, encryption function $E(\cdot)$.

**Eusure:** $Z_T^s$

1: $m^e = E(m, k_1)$
2: $l = Len(m^e)$
3: **for** $k \in \{1, 2, \cdots, l\}$ **do**
4:      **for** $u \in \{0, 1, \cdots, 6\}$ **do**
5:          **for** $i \in \left\{1, 2, \cdots, H^{'} \times W^{'} \times C^{'}\right\}$ **do**
6:              **if** $D[i] \in MS_u$ **then**
7:                  $z_i \sim \mathcal{N}\left(\mu_{T-1}^i, (\sigma_{T-1}^i)^2\right)$
8:                  $pool_{k,0} = \left(-\infty, \mu_{T-1}^i - \gamma\right)$
9:                  $pool_{k,1} = \left(\mu_{T-1}^i + \gamma, +\infty\right)$
10:                  $(Ind[0], Ind[1]) = E(k_2, pool_{k,0}, pool_{k,1})$
11:                  $z_i^s = trunc\left(\mu_{T-1}^i, \sigma_{T-1}^i, \left(c\left[m_k^e\right][0], c\left[m_k^e\right][1]\right)\right)$
12:              **end if**
13:          **end for**
14:      **end for**
15: **end for**

---

using the shared key $k_2$. The encrypted candidate pools are labeled $Ind[0]$ and $Ind[1]$ from left to right, respectively. When $z_i^{s'} \leq \mu_{T-1}$, $m_k^{e'} = 0$. When $z_i^{s'} > \mu_{T-1}$, $m_k^{e'} = 1$. The above operation is repeated until all the secret data $m^{e'}$ are extracted. Finally, the secret data $m^{e'}$ is decrypted using the key $k_1$ to obtain $m^{'}$.

$$\begin{cases} m_k^{e'} = 0, & if\, z_i^{s'} \leq \mu_{T-1} \\ m_k^{e'} = 1, & if\, z_i^{s'} > \mu_{T-1} \end{cases} \tag{13}$$

## 4 EXPERIMENTAL RESULTS

In our experiment, we chose a publicly pretrained Latent Diffusion Model[1] to perform the generative image steganography task, where the size of the latent space is $32 \times 32 \times 4$ and $r = 0.3$. The forward diffusion and backward processes all consisted of 50 steps. All experiments were executed on a GeForce RTX 1080Ti, and LDStega requires no additional training or fine-tuning of the LDM. We assess the flexibility of LDStega's capacity by varying the conditions of the targeted steganography positions. To demonstrate the controllability of LDStega, LDStega designed experiments in two scenarios based on the condition text. To demonstrate the superiority of LDStega in terms of availability and robustness, we compare it with two steganography with embedding (SE) methods, namely Hidden [42], CHAT-GAN [28] and four state-of-the-art (SOTA) GIS methods, namely IDEAS [19], S2IRT [41], and StegaDDPM [23]. The above methods were evaluated on Bedroom and Cat datasets [37], as well as on face images from FFHQ [15]. LDStega categorizes the dataset into three classes, namely human faces, animals, and general objects (such as bedrooms, architecture, etc.). Extraction accuracy (Acc) and capacity are used to evaluate the decoding accuracy of secret data and steganographic capacity, respectively.

---

[1] https://github.com/CompVis/latent-diffusion

**Table 1: The comparison of steganographic capacity and extraction accuracy for three distinct scenarios.**

| Position | Capacity (bits) | File types | LDStega | | |
|---|---|---|---|---|---|
| | | | FFHQ | Bedroom | Cat |
| $(MS_0)$ | 1636 | PNG | 99.51% | 99.32% | 99.46% |
| | 1129 | JPEG | 98.34% | 98.27% | 98.22% |
| $(MS_0, MS_1)$ | 2805 | PNG | 99.22% | 99.39% | 99.28% |
| | 2085 | JPEG | 97.96% | 98.01% | 98.11% |
| $(MS_0, \cdots, MS_6)$ | 4096 | PNG | 98.65% | 98.50% | 98.48% |
| | 4096 | JPEG | 96.15% | 95.42% | 96.28% |

### 4.1 Flexibility of steganographic capacity

To validate the effectiveness of LDStega's hiding strategy in the case of low steganographic capacity, which selects targeting positions where $D$ exhibits relatively low values to hiding secret data, Table 1 presents the steganographic capacity and extraction accuracy for three distinct conditions of the targeted steganography positions: $(MS_0), (MS_0, MS_1), (MS_0, \cdots, MS_6)$. By employing the mapping function to hide secret data within $Z_T^s$ at the specified position $(MS_0)$ in $D$, we achieve a steganographic capacity of 1639 bits for the PNG format, with outstanding extraction accuracies of 99.51%, 99.32%, and 99.46% for the FFHQ, Bedroom, and Cat datasets, respectively. Transitioning to JPEG format slightly reduces the steganographic capacity to 1129 bits, but the extraction accuracy is still commendable, at 98.34%, 98.27%, and 98.22% for the respective datasets. When focusing on positions $(MS_0, MS_1)$ in $D$ for $Z_T^s$, the PNG format delivers a higher steganographic capacity of 2805 bits, accompanied by equally impressive extraction accuracies of 99.22%, 99.39%, and 99.28% for the FFHQ, Bedroom, and Cat datasets. In the case of JPEG format, a steganographic capacity of 2085 bits is observed, along with strong extraction accuracies of 97.96%, 98.01%, and 98.11% for the FFHQ, Bedroom, and Cat datasets. Table 1 confirms that LDStega's position selection strategy effectively optimizes steganographic performance under low steganographic capacity, while also maintaining a high extraction accuracy even if the capacity is 4096 bits.

### 4.2 Practicality

*4.2.1 Controllability.* To verify the controllability of our proposed LDStega with regard to image content generation, as illustrated in Fig. 5, LDStega presents the generated stego images for two scenarios based on the input condition text. In the first scenario, the sender generates personalized stego images for different scenes using distinct condition text. For instance, when inputting descriptions such as "Mickey Mouse and Donald Duck" or "Van Gogh's Starry Night" into the model, LDStega will generate the corresponding stego images. In the second scenario, the sender generates multiple stego images for the same scene, all based on the same condition text. For example, when "A vase brimming with flowers" and "A cactus that grows in the desert" are entered, the second row of Fig. 5 displays four stego images corresponding to condition text. LDStega excels at tailoring steganographic images to align with the sender's characteristics, including their occupation and interests, thereby preventing any suspicion from arising due to

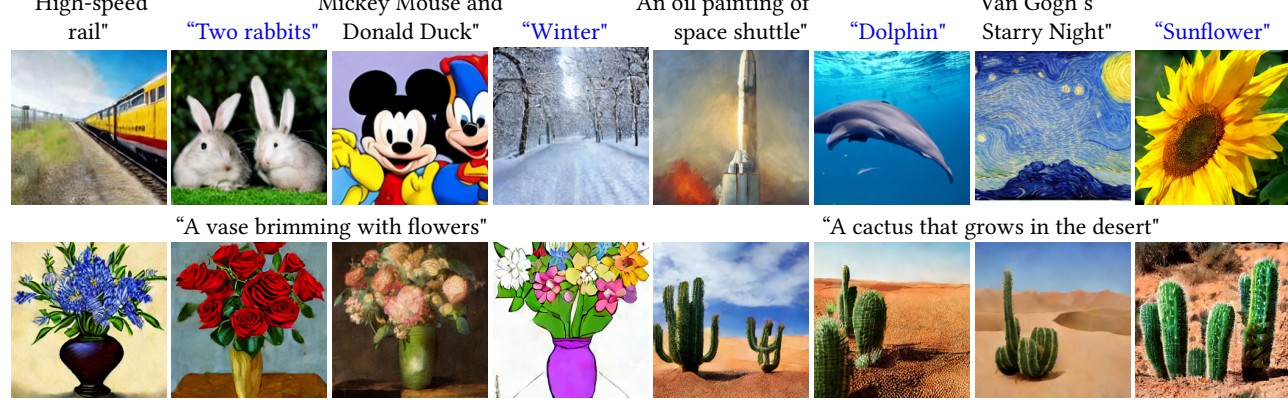

"High-speed rail"    "Two rabbits"    "Mickey Mouse and Donald Duck"    "Winter"    "An oil painting of space shuttle"    "Dolphin"    "Van Gogh's Starry Night"    "Sunflower"

"A vase brimming with flowers"    "A cactus that grows in the desert"

**Figure 5: The stego images generated by LDStega are dependent on the input text. In the first row, the sender creates personalized stego images for different scenes using distinct condition text. In the second row, the sender generates multiple stego images for the same scene, where the four leftmost columns and the four rightmost columns are each generated from the same text.**

**Table 2: Performance comparison of LDStega with SOTA work in terms of extraction accuracy, capacity, and generated image size when the stego image is saved in both PNG and JPEG formats, respectively.**

| File type | Types | Approches | Acc (%) | | | Capacity (bits) | Image size |
|---|---|---|---|---|---|---|---|
| | | | FFHQ | Bedroom | Cat | | |
| PNG | SE | Hidden [42] | 62.40% | 62.06% | 62.09% | 256 | 128 × 128 |
| | | CHAT-GAN [28] | 91.08% | 90.83% | 92.71% | 4096 | 256 × 256 |
| | GIS | IDEAS [19] | 70.16% | 69.35% | 70.56% | 2048 | 256 × 256 |
| | | S2IRT [41] | 70.89% | 70.87% | 70.91% | 4096 | 64 × 64 |
| | | StegaDDPM [23] | 93.45% | 90.19% | 90.81% | 4096 | 256 × 256 |
| | | LDStega | **98.65%** | **98.50%** | **98.48%** | 4096 | 256 × 256 |
| JPEG | SE | Hidden [42] | 62.02% | 61.01% | 61.67% | 256 | 128 ×128 |
| | | CHAT-GAN [28] | 50.09% | 50.56% | 49.93% | 4096 | 256 × 256 |
| | GIS | IDEAS [19] | 60.23% | 59.01% | 50.21% | 2048 | 256 × 256 |
| | | S2IRT [41] | 70.12% | 70.23% | 70.17% | 4096 | 64 × 64 |
| | | StegaDDPM [23] | 51.65% | 51.21% | 51.19% | 4096 | 256 × 256 |
| | | LDStega | **96.15%** | **95.42%** | **96.28%** | 4096 | 256 × 256 |

unusual behavior. In contrast, methods such as IDEAS [19], S2IRT [41], StegaDDPM [23], RoSteALS [4] fall short in this aspect. These techniques are limited to generating steganographic image content within the boundaries of their training dataset, lacking autonomous control over features such as style, object count, and color in the generated images. Given that our proposed approach grants precise control over the content of generated images, potential attackers are unable to discern the presence of steganography by scrutinizing the application context of steganographic images, thus enhancing the security of steganographic practices.

*4.2.2 Availability.* To improve the availability of steganography in realistic scenarios, we discretized the pixel values of the stego images into integers and subsequently saved them in PNG and JPEG formats. As shown in Table 2, we compare our work with two types of steganography methods in terms of steganographic capacity, extraction accuracy, and stego image size: 1) SE methods based on deep learning, in which the cover image is modified to perform secret data embedding (including Hidden [42] and CHAT-GAN [28],

where Hidden [42] is trained without a noise layer). 2) GIS methods (including IDEAS [19], S2IRT [41], and StegaDDPM [23]). The results in Table 2 demonstrate that LDStega outperforms the SOTA methods in both PNG and JPEG formats.

Specifically, when the stego image is PNG format, Hidden [42], IDEAS [19], and S2IRT [41] exhibit significantly lower extraction accuracy of secret data at the corresponding capacity, which can not meet the communicating parties' requirements for information accuracy. While CHAT-GAN [28], StegaDDPM [23], and LDStega perform relatively well. Notably, LDStega achieves impressive extraction accuracy of 98.65%, 98.50%, and 98.48% on FFHQ, Bedroom, and Cat datasets, respectively. The experiments indicate that Hidden [42] and S2IRT [41] exhibit weak resistance to quantization errors, whereas CHAT-GAN [28], StegaDDPM [23] and LDStega demonstrate robustness against quantization errors. Additionally, it was observed that the low extraction accuracy of IDEAS [19] is attributed to its steganographic capacity rather than quantization errors. Even when the stego image is in JPEG format, LDStega maintains a consistently high extraction accuracy that remains

**Table 3: Comparison of the extraction accuracy of different methods against different types of image attacks when saving the stego image in PNG format on the FFHQ, Bedroom, and Cat datasets.**

| Attack | Factor | CHAT-GAN [28] | | | StegaDDPM [23] | | | LDStega | | |
|---|---|---|---|---|---|---|---|---|---|---|
| | | FFHQ | Bedroom | Cat | FFHQ | Bedroom | Cat | FFHQ | Bedroom | Cat |
| Identity | - | 91.08% | 90.83% | 92.71% | 93.45% | 90.19% | 90.81% | **98.65%** | **98.50%** | **98.48%** |
| Crop | $p = 0.95$ | 86.58% | 87.56% | 88.55% | 88.89% | 86.23% | 86.72% | **89.77%** | **90.26%** | **90.83%** |
| Salt & Pepper noise | $\varrho = 0.01\%$ | 90.90% | 90.73% | 91.81% | 93.24% | 90.15% | 90.66% | **97.98%** | **97.76%** | **97.73%** |
| | $\varrho = 0.04\%$ | 90.38% | 89.99% | 91.05% | 93.21% | 90.10% | 90.36% | **95.64%** | **94.69%** | **95.70%** |
| | $\varrho = 0.07\%$ | 89.48% | 88.45% | 90.29% | 93.17% | 90.03% | 90.24% | **93.58%** | **92.55%** | **94.22%** |
| Gaussion noise | $\varrho = 0.01\%$ | 90.35% | 90.51% | 92.04% | 77.15% | 71.01% | 71.86% | **98.06%** | **97.63%** | **98.20%** |
| | $\varrho = 0.04\%$ | 90.15% | 89.46% | 91.91% | 66.59% | 61.77% | 62.78% | **96.14%** | **92.78%** | **96.37%** |
| | $\varrho = 0.07\%$ | 89.91% | 88.87% | 90.19% | 63.10% | 59.07% | 59.80% | **92.64%** | **89.02%** | **94.45%** |
| JPEG compression | $Q = 90$ | 50.31% | 50.40% | 50.49% | 55.89% | 53.90% | 53.54% | **98.28%** | **98.18%** | **98.27%** |
| | $Q = 70$ | 50.25% | 50.34% | 50.47% | 52.10% | 52.20% | 51.39% | **96.68%** | **96.95%** | **97.31%** |
| | $Q = 50$ | 50.17% | 50.29% | 50.41% | 51.24% | 51.37% | 50.94% | **94.29%** | **94.46%** | **95.58%** |

**Table 4: The extraction accuracy of LDStega against different types of attacks when saving the stego image as JPEG format.**

| Attack | Factor | LDStega | | |
|---|---|---|---|---|
| | | FFHQ | Bedroom | Cat |
| Identity | - | 96.15% | 95.42% | 96.28% |
| Crop | $p = 0.95$ | 87.76% | 86.84% | 88.50% |
| Salt & Pepper noise | $\varrho = 0.01\%$ | 95.43% | 93.85% | 95.89% |
| | $\varrho = 0.04\%$ | 93.23% | 90.87% | 94.35% |
| | $\varrho = 0.07\%$ | 91.24% | 87.91% | 93.23% |
| Gaussion noise | $\varrho = 0.01\%$ | 95.79% | 93.36% | 96.12% |
| | $\varrho = 0.04\%$ | 93.41% | 87.34% | 94.42% |
| | $\varrho = 0.07\%$ | 90.53% | 83.33% | 93.06% |
| JPEG compression | $Q = 90$ | 96.12% | 95.25% | 96.20% |
| | $Q = 70$ | 95.60% | 94.09% | 95.76% |
| | $Q = 50$ | 90.53% | 88.08% | 92.54% |

unattainable by two types of steganography methods. Furthermore, our observations reveal that stego images with the PNG format lead to a higher extraction accuracy of secret data compared to the JPEG format. This is attributed to JPEG's use of lossy compression for reducing image file size, whereas PNG utilizes lossless compression, making it more suitable for preserving the integrity of hidden information. In conclusion, LDStega emerges as an efficient and competitive GIS method with high availability in realistic scenarios.

### 4.3 Robustness

To evaluate the robustness of LDStega, we subjected the generated stego images to commonly used image attacks, including Identity, Crop, Salt & Pepper noise, Gaussian noise, and JPEG compression. These attacks are known to increase the difficulty of extracting secret data. We focus on analyzing and discussing the robustness of LDStega's PNG and JPEG stego images against these attacks. Table 2 demonstrates that Acc of Hidden [42], IDEAS [19], and S2IRT [41] are already considerably low, even in the absence of attacks. Therefore, when the stego image is PNG format, Table 3 compares Acc of LDStega with CHAT-GAN [28] and StegaDDPM [23] under the aforementioned attacks. The results in Table 3 illustrate that

LDStega exhibits excellent adaptability to various common image attacks, maintaining a high extraction accuracy for secret data. In particular, when resisting JPEG compression, the Acc of CHAT-GAN [28] and StegaDDPM [23] is close to 50%, whereas LDStega continues to uphold a high extraction accuracy. The main reason is that the secret data are encoded in the latent space of generated images. LDM lies in its natural robustness to noise and perturbations due to its inherent Gaussian noise characteristics of latent space. Thus, combined with the robust mapping function, we can still achieve high decoding accuracy for secret data in the case where the stego image undergoes discrete quantization and lossy channels. When the stego image is in JPEG format, Table 4 only displays the experimental results of the proposed methods under the attacks of Identity, Crop, Salt & Pepper noise, Gaussian noise, and JPEG compression. This is because CHAT-GAN [28] and StegaDDPM [23] have already shown low resistance to attacks in the PNG format, failing to meet the basic requirements of steganography. It can be observed from Table 4 that LDStega still maintains a high Acc in JPEG format, which proves the superiority of LDStega.

## 5 CONCLUSION

In this paper, we propose LDStega, a practical and robust generative image steganography based on LDM. By reconstructing the latent space and applying quantization and noise to the generated image, we draw three conclusions about LDM, and based on three conclusions we design a coding strategy to hide secret data on the latent space, achieving controllable, available, and robust generative image steganography. The controllability of LDStega allows senders to generate personalized stego images for different scenarios with the sender's characteristics, including their occupation and interests, thereby preventing any suspicion from arising due to unusual behavior. Furthermore, we compare LDStega with the SOTA methods. Experimental results demonstrate that LDStega maintains a high extraction accuracy when stego images are saved in the widely used PNG and JPEG formats. LDStega also exhibits superior resistance to common image attacks compared to existing techniques. Consequently, LDStega shed light on the practical application of generative image steganography in real-world scenarios.

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
