# OpenReview forum: "LDStega: Practical and Robust Generative Image Steganography based on  Latent Diffusion Models"
_acmmm.org/ACMMM/2024/Conference — MM2024 Poster_

### Official Review · Reviewer_SzPW · 2024-05-09

**Rating:** 4
**Confidence:** 3

**Summary:**

This paper proposes the LDStega method. Experimental results show that LDStega can control the content of generated images while also saving steganographic images in PNG and JPEG formats, achieving high extraction accuracy.

**Strengths:**

This paper discusses an interesting area and achieves superior performance than existing methods.

**Limitations:**

Several limitations are shown as follows:

(1) How stable is the method? That is, the experimental results lack standard deviation.

(2) The fonts in Figure 2 are not uniform and the modules are not aligned. Please modify them carefully to improve the professionalism of the drawing.

(3) Does the performance of the proposed method mainly come from diffusion or what? This aspect must be experimentally compared and discussed.

If the above questions are adequately answered, I may consider increasing my existing score.

**Suitability:**

3

---

### Official Review · Reviewer_gDSz · 2024-05-22

**Rating:** 5
**Confidence:** 4

**Summary:**

This paper proposes LDStega, a practical and robust generative image steganography based on LDM. By reconstructing the latent space and applying quantization and noise to the generated image, LDStega designed a coding strategy to hide secret data on the latent space, achieving controllable, available, and robust generative image steganography.

**Strengths:**

1.	LDStega can conceal secret data within the noise and utilizes a deterministic sampler to generate the stego latent space. Notably, LDStega does not require fine-tuning pre-trained models or training additional models.

2.	LDStega utilizes the controlled condition text as input, enabling the sender to generate personalized stego images for various scenarios with the sender’s characteristics, thereby avoiding suspicion arising from unusual behavior

3.	LDStega design a coding strategy that selects a sampling interval from a candidate pool of truncated Gaussian distributions guided by secret data to generate the stego latent space, which ensures high extraction accuracy in resisting common image attacks.

**Limitations:**

1.	LDStega consistently applies a truncation interval of 0.3 for r. Can r be reduced below 0.3 in low-capacity steganographic scenarios, thereby enhancing the quality of the stego images without compromising the high extraction rate?

2.	Given that Pulsar [1] is another image steganography algorithm based on diffusion models, it merits inclusion in the reference section to ensure a thorough literature review.
[1] Jois T M, Beck G, Kaptchuk G. Pulsar: Secure Steganography through Diffusion Models[J]. Cryptology ePrint Archive, 2023.

3.	The manuscript encompasses a multitude of variables; a thorough examination is required to confirm the proper utilization of each variable name.

4.	The experimental setup in Section 4.1 should be revised, as the title "Flexibility of Steganographic Capacity" is not consistent with the purpose of the experiments discussed.

**Suitability:**

3

---

### Official Review · Reviewer_snrz · 2024-05-24

**Rating:** 5
**Confidence:** 4

**Summary:**

The proposed method LDStega utilizes large-scale LDM communities that have contributed an extensive collection of freely available open-source tools. LDStega has a broader range of applications and offers a new reference pattern for steganography design.

**Strengths:**

LDStega utilizes the controlled condition text as input, enabling the sender to generate personalized stego images for various scenarios with the sender’s characteristics, thereby avoiding suspicion arising from unusual behavior.  Furthermore, LDStega saves stego images in the commonly used PNG and JPEG formats, making this method more practical.

**Limitations:**

1. LDStega leverages a pre-trained Latent Diffusion Model to hide the secret data in the latent space. However, the transformation of the stego image into the latent space by the receiver will incur information loss. Why not consider training a lightweight secret data extractor for extraction.
2. The experimental design in Table 1 requires refinement. To validate the effectiveness of LDStega's hiding strategy in the case of low steganographic capacity, a comparative analysis should be conducted. This would involve assessing the accuracy of extraction when employing LDStega's strategy at a low embedding capacity, as opposed to the accuracy achieved without utilizing LDStega under same low steganographic capacity conditions.
3. The English grammar, spelling, and sentence structure of this paper should be improved, so that the goals and results of the study are clearer to the reader.
4. For improved readability, the numerical clarity in both the horizontal and vertical axes of Figure 4 should be enhanced.
5. LDStega utilizes a truncated Gaussian distribution for hiding to improve the extraction rate of secret data. A comprehensive analysis is needed to identify the elements within the LDStega framework that contribute to potential decoding losses of the secret data.

**Suitability:**

2

---

### Official Review · Reviewer_pgJE · 2024-06-02

**Rating:** 2
**Confidence:** 4

**Summary:**

This paper introduces a robust generative steganographic scheme based on the LDM.  The scheme proposes a mapping method to hide binary secret data, applying a truncated Gaussian distribution sampling method.   It also takes the text as a condition to guide the generation of stego images.  Though better performances are achieved compared to existing methods, the paper has some weaknesses.

**Strengths:**

This paper proposes a new mapping function to hide secret data during the diffusion process of LDM, which is well organized. For instance, it introduces the principle of diffusion models, counting the number of elements in six different intervals and showing the algorithms and results of different text prompts in the paper. At last, it demonstrates detailed experiments on various datasets.

**Limitations:**

However, some weaknesses exist in the paper.
1)  the novelty of this paper is limited. This paper only proposes a mapping function, which is improved slightly on existing methods. It seems the paper is not innovative.
2) Steganalysis is not conducted in the paper, which is necessary for steganography.
3) Some existing comparative methods, like work[1], are not included in the paper. They also demonstrate good robustness.
4) The experiments do not verify the so-called  "practical and robustness steganography".  The noising factors in Table 3 and Table 4 are too small, which seems impractical in reality
5) This paper is not clearly written, and the secret mapping process is perplexing. For example, line 11 in Algorithm 2 is not elaborated, where $Ind[0]$ is the same as $C[.][0]$?
6) Do different text prompts affect the performances?

[1] Wei P, Zhou Q, Wang Z, et al. Generative steganography diffusion[J]. arXiv preprint arXiv:2305.03472, 2023.

**Suitability:**

2

---

### Meta-Review · Program_Chairs · 2024-07-04

**Recommendation:** Accept (Poster)
**Confidence:** 4

**Metareview:**

The manuscript has been reviewed by four reviewers, and has received mixed reviews post rebuttal,

The comments from Reviewer pgJE, who is an experienced expert in the field, gave detailed comments and in-depth analysis. After the rebuttal, the reviewer is not convinced by the authors' response, including the comments on the novelty, among others. There are also some concerns on, for example, the Steganalysis.

Since MM seeks the highest quality, the manuscript in the current form, unfortunately, could not be accepted. The AC understands that this might be a sad news, and hopes the authors could address the major concerns raised by the reviewers in the next version.

***TPC Addendum***
Besides official reviews, this paper was discussed between TPC, SAC, and AC. The limitations on novelty were noted but we also discussed the importance of the application area and the strengths in terms of unified methodology and personalization. Three of the four reviewers recommended acceptance, and two also raised their scores after rebuttal. While the paper is very close to the borderline, we are suggesting a poster acceptance to allow for the debate and discussion to continue at the conference.